# Seismic Performance of Precast Concrete Frame Beam-Column Connections with High-Strength Bars

**DOI:** 10.3390/ma15207127

**Published:** 2022-10-13

**Authors:** Jianbing Yu, Ershuai Zhang, Zhiqiang Xu, Zhengxing Guo

**Affiliations:** 1College of Civil Science and Engineering, Yangzhou University, Yangzhou 225127, China; 2School of Civil Engineering, Southeast University, Nanjing 210096, China

**Keywords:** precast concrete, beam-to-column connection, seismic behavior, finite element

## Abstract

As the construction industry is striding towards the industrialization of green buildings, a precast concrete frame beam-column joint with high-strength reinforcement was proposed. Simulate reversed cyclic loading was carried out on two precast connections and one cast-in-place connection to examine the seismic behavior of the proposed new precast connection. The main test variables between the two precast connections were the strength of the reinforcement at the bottom of the beam. The failure shape, hysteresis curve, skeleton curve, strength, deformation ability, stiffness degradation, and energy dissipation were monitored and compared with the cast-in-place connection. The findings of this paper showed that the precast joints had good strength reserve, and the seismic performance in the later stage of loading even exceeds the cast-in-place joints. It was also found that the plastic hinge zone of the beam could be moved away from the column surface via reinforcement method. Additionally, based on the experiment, a detailed nonlinear finite element analysis (FEA) method was developed to reproduce the test response of specific types of bending moment-resistant precast concrete beam-column connections under a reversed loading test, which provided a theoretical reference for further research of the connections.

## 1. Introduction

The application of concrete has a long history, and the research on concrete structures has never ceased, especially regarding concrete beams, columns, walls and slabs [1]. However, with the increasing awareness of environmental protection and the aging of on-site workers, conventional cast-in-place concrete may be replaced by other construction methods shortly due to its labor-intensive and resource-intensive characteristics. The use of precast concrete structural systems can reduce the environmental impact, and this type of construction can effectively reduce the labor force on the construction site and solve the problem of labor aging [2,3,4,5]. The precast concrete frame structure has the advantages of a flexible building layout, easy standardization of beams, columns, and other components, high prefabrication rate, and wide application range. For example, some large open space factories, shopping malls, parking lots, office buildings, teaching buildings, hospitals, and other buildings can adopt precast frame structures, which is a structural system that is widely studied and applied in precast structures [6]. A precast frame structure refers to prefabricated columns and beams which are processed in a precast factory, transported to the construction site, and then assembled to form an overall structure [7]. However, the panel zone of the precast concrete frame is often a weak part of the structure. The connection of beams and columns directly affects the stability and seismic resistance of the overall structure. Therefore, ensuring that the panel zone of the precast frame structure has good mechanical properties is of great significance to the safety of the overall structure [8,9]. Since the 1990s, various new connection types suitable for use in seismic areas have been proposed for precast structures, and a series of experimental studies and theoretical analyses have been performed [10]. At present, the most widely used and researched type of connections are the wet connections, also known as the prefabricated post-cast integral frame beam-column connection. The research mainly focuses on the related properties of its connecting steel bar, steel connecting form and post-cast concrete, and finally puts forward a series of measures to effectively improve the performance of beam-column connections [11].

A large number of scholars have proposed post-cast monolithic prefabricated concrete frame joint systems with reinforced anchorage or lap connection. Im et al. [12] proposed a prefabricated beam-column joint using U-shaped beam shells. The research results showed that the seismic performance of the fabricated joints was equivalent to that of the cast-in-place joint, but its stiffness and energy absorption capacity were poor. Parastesh et al. [13] proposed a new type of precast ductile flexural beam-column joint for high-intensity seismic areas. The research results showed that the joint system had better ductility and energy absorption capacity than cast-in-place joints. Wu et al. [14] designed a new type of frame beam-column rebar lapped joint. The study found that the joint had a similar seismic capacity to the cast-in-place concrete joint, but its bearing capacity degradation rate was significantly faster than that of the cast-in-place joint. Restrepo et al. [15] proposed a beam-column joint in which the precast beam bottom reinforcement extended into the core area of the joint and was anchored by a 90° hook. It was found that the concrete is not compacted due to the crowding of steel bars in the core area of the joint. The seismic performance of a new post-cast monolithic joint was studied by Guan et al. [16]. The results showed that the proposed joint had good hysteretic behavior and energy dissipation capacity. Einea et al. [17], Alias et al. [18], and Ling et al. [19] studied the effects of sleeve diameter, steel anchorage length, and inner wall structure on the bond strength of sleeve grouting connectors. The test results indicated that with the decrease of the diameter of the sleeve, the tensile strength of the rebars increased, and the stiffness and ductility also increased. The longer the anchorage length of the rebars, the stronger the bonding strength of the sleeve connector. Through the above research, it was found that for precast concrete connectors with anchor longitudinal bars, because of the large number of anchoring straight bars, the panel zone was crowded, and the working space of the post-casting node was narrow, which greatly increases the difficulty of setting stirrups and pouring concrete, and it was difficult to guarantee the construction quality.

To solve the problem that steel bars were prone to collision in the panel zone, some scholars had proposed a series of measures to enhance the connection performance. Gou et al. [20] proposed the use of low-shrinkage engineering cementitious composites (LSECC) as post-cast concrete beam-columns in the panel zone. The use of cement-based composites could reduce the density of reinforcement in the joint area without affecting its seismic performance. Lee et al. [21] proposed a beam-column joint with an anchor head on the bottom rebar of the beam. The study found that the seismic performance of the joint was equivalent to that of the cast-in-place joint, but because the bottom bar of the beam has an anchor head, it will increase the possibility of cracking in the panel zone. Ha et al. [22] designed a U-shaped steel strand semi-precast concrete beam-column joint suitable for fortified earthquakes. The seismic performance of the new prefabricated joint could meet the requirements for use in moderate earthquake areas. Yu et al. [23] proposed steel strands anchored precast concrete beam-column joints. The test found that the joint system had good seismic performance, but due to the unbounded section of the steel strand at the keyway, the concrete spalling was serious in this part. Zhao et al. [24] proposed post-cast monolithic beam-column joints of prefabricated high-strength concrete and post-cast monolithic joints of high-strength steel fiber reinforced concrete. The study found that pouring steel fiber reinforced concrete in the post-cast area of the prefabricated structure could not only reduce the number of stirrups in the node core area and facilitate construction, but also improve the cracking strength of the joint and the ductility and energy absorption capacity of the joint. To sum up, the small-diameter high-strength steel bars have a short anchorage length and required little space for bending. The panel zone of post-cast joints with special materials can replace part of the stirrups to bear the shear force. The two improvement methods can solve the problem of excessively dense reinforcement in the panel zone, which not only facilitate construction but also improve the quality of on-site pouring. The overall performance is the same as that of the cast-in-place joints, and it could be used in fortified earthquake areas.

This paper presents an assembled concrete frame beam-column joint with high-strength reinforcement. The small-diameter high-strength bars are used to replace ordinary steel bars at the bottom of the prefabricated beam. When assembling the prefabricated components, the small-diameter steel bars can move freely within the range of the keyway to avoid collision between the tensile bars of beams and columns. Additionally, to avoid the premature failure of the concrete in the keyway of the prefabricated beam with high strength reinforcement under loading, the core beam is set in the keyway. Besides, in the context of global initiatives for energy conservation and emission reduction, it has become a general trend to promote the future use of high-strength materials in the construction industry. The application of high-strength materials can improve the service life of a structure, reduce the weight of a structure, reduce the dense configuration of steel bars, facilitate construction, ensure project quality, and promote energy conservation and environmental protection.

## 2. Experimental Program

### 2.1. Design and Details of the Specimens

The connection design was improved based on the reference by Yu et al. [23]. Through the research, it was found that the precast joints lost their working ability because the key groove at the bottom of the precast beam was crushed in advance. Therefore, the two precast connections tested in this paper were designed as a core beam inside the keyway to prevent the concrete from being crushed. 

The specimens were tested, including a full-scale cast-in-place specimen and two full-scale precast specimens. The cast-in-place specimen was labeled as XJ, as shown in Figure 1. The length of the beams on both sides of the column was 2.0 m. Closed stirrups of B8 reinforcement bars were placed in the beam at a spacing of 100 mm from the column surface and 200 mm from both ends of the beam. Five C20 bars were placed at top of the beams, and three bars were placed at the bottom of the beams. The total height of the column was 2.825 m. Four C25 and eight C20 bars were placed in the column. The column stirrup spacing was 100 mm within the panel zone and within the range of 600 mm at the top and bottom of the panel zone. The stirrups in the other parts of the column were B10@200. The ends of the column and the beam were wrapped with a rectangular steel hoop to prevent local damage during the test. Precast beams and precast columns were assembled into a precast connection, and the precast column reinforcement and cross-section of the two precast connections were the same, as shown in Figure 2. The lengths of the bottom-column bars protruding from the upper surface of the column were 670 mm and 630 mm, respectively. The precast beams of YZ1 and YZ2 specimens are shown in Figure 3 and Figure 4. The section height of the precast beam was 550 mm, which consisted of a 330 mm precast beam and a 120 mm laminated layer. The bottom force bars of the YZ1 precast beam extend 450 mm outwards and bend upwards by 200 mm. The end of the strands at the bottom of the YZ2 precast beam adopted embossed anchors to enhance the new anchoring performance between the strands and the concrete. The precast beams of the two prefabricated connections were provided with a keyway with a length of 600 mm at the beam end, while the thickness of the sidewall of the keyway was 50 mm, and the thickness of the bottom wall was 65 mm. Two C10 steel bars were used in the keyway at 150 mm from the bottom of the beam to facilitate the fixation of the core beam stirrup. More details can be found in Figure 5 and Figure 6.

### 2.2. Construction Process

All specimens were processed and manufactured in a precast component factory. The XJ connection was poured in a flat-laying manner. First, the steel cage was bound. Then, the formwork was supported, and, finally, the concrete was poured. After the pour was completed, it would be cured under natural conditions. The production process of precast connections is as follows: The first step was to make the precast components as shown in Figure 7c. Secondly, after the strength of the precast components met the requirements, the prefabricated components were assembled. The upper and lower columns were connected and fixed first, and then the precast beams were put in place. After all the precast components were adjusted in place, the panel zone concrete, the precast beam keyway, and the laminated concrete were poured. Finally, the grouting sleeves of the precast column were grouted to complete the assembly work of the precast beam-column connections. The relevant production process is shown in Figure 7.

### 2.3. Materials Properties

Three cubes were made for each batch of concrete, and then they were cured under the same conditions as the full-scale specimens. Three samples were taken for each kind of steel bar for direct tensile tests. The mechanical properties of concrete and rebars are shown in Table 1 and Table 2.

### 2.4. Test Setup and Loading Procedure

In this test, a 1500 kN hydraulic servo control system (MTS) was used to apply a horizontal load on the top of the column. The axial load of the column was applied to the steel strand through a through-hole jack. The axial pressure ratio was controlled to 0.27, corresponding to the axial load carrying capacity. The loading device is shown in Figure 8.

For quasi-static loading tests, Chinese code recommends the use of a load-deformation dual-control loading system (JGJ/T 101, 2015) [25]. However, human subjectivity often plays a big role in the conversion of load to deformation. This loading is according to the used drift ratio control (ACI 374.1-05, 2005) [26]. The drift ratio is the ratio of the actuator loading displacement to the distance of the inflection point of the column. Before the formal loading, pre-apply a displacement of 1–2 mm to the specimens. The drift ratios were 0.2%, 0.25%, 0.35%, 0.5%, 0.75%, 1%, 1.5%, 2%, 2.75%, 3.5%, and 4.25%. Each cycle was repeated three times, as shown in Figure 9.

## 3. Experimental Results and Analysis

### 3.1. Crack Pattern and Failure Mode

There were no cracks that appeared during preloading with a drift ratio of 0.034% and 0.069%. The specimen was loaded with a 0.2% drift ratio for the first stage, and several cracks at the bottom of the beam appeared at approximately 5 cm, 10 cm, 35 cm, and 62 cm from the beam-column joint surface. When loading at a drift ratio of 0.35%, a crack with a width of 0.06 mm at the top of the beam appeared at approximately 55 cm from the beam-column joint surface. When loaded at a 0.5% drift ratio, there were many vertical bending cracks at the top of the beam, and the cracks in the upper and lower beams penetrated each other. When loaded to a 0.75% drift ratio, visible cracks appeared at the beam-column junction with a width of 0.6 mm. As the loading drift ratio increased, the width, length, and number of cracks increased. The distribution of the cracks gradually expanded toward the loading end, and the number of diagonal cracks increased. When loaded at a drift ratio of 4.25%, the concrete at the lower end of the beam experienced substantial spalling. It was observed that the lower longitudinal rebar was bent when it was compressed. At a distance of 10 cm from the beam-column interface, a wide main crack formed in the middle of the beam, and the load dropped significantly, lower than 80% of the peak load, and the loading was complete. The final failure pattern of specimen XJ is shown in Figure 10a.

First, we carried a single load of 0.034%, 0.069%, and 0.10% drift ratio on YZ1. When the 0.069% drift ratio was loaded, a crack with a width of 0.02 mm appeared in the lower part of the beam at a distance of 11 cm from the beam-column interface. When loaded to a drift ratio of 0.10%, bending cracks that developed from the bottom to top appeared at the bottom of the beam at 6 cm, 38 cm, and 60 cm from the beam-column interface, respectively. After loading to the drift ratio three times per stage, the cracks gradually developed until the specimens were damaged. When loading at a 4.25% drift ratio, the concrete at the beam end was continuously crushed, and the lower concrete continuously fell. It could be observed that while the lower longitudinal reinforcement bends when compressed, the concrete inside the core beam was also crushed, and the above-mentioned diagonal cracks continued to widen, forming a major diagonal crack. When the 4.25% drift ratio was repeatedly loaded for the second time, the load value was significantly reduced, the specimen was damaged, and the loading was stopped.

The specimen YZ2 failure pattern was almost similar to the YZ1 specimen. When loaded to a drift ratio of 3.5%, a large amount of concrete under the beam spalled off. The steel strand was exposed, the strand came loose when pressed, and the stiffeners fractured. When the third loading was carried out at a drift ratio of 3.5%, the load dropped to 80% of the peak value, and the testing was ceased.

Through the low-cycle repeated loads test process of three joints, it was found that all specimens had experienced the elastic stage, crack propagation stage, and failure stage. During the loading process, the cracking load of the cast-in-situ joint was higher than that of precast joints, because the precast joints had the new and old concrete interface, cracks appeared at this place during the loading process, and they would soon penetrate up and down. In general, the expected beam end bending failure occurred at the three joints, and only a few small cross-oblique cracks appeared in the joint core area.

### 3.2. Hysteresis Characteristics

The hysteresis curve shows the deformation process of the component, which can be used to fully understand the characteristics of stiffness, strength, deformation, energy absorption, and damage to the component of a structure. The hysteresis curves of the three specimens are shown in Figure 11.
(1)It can be seen that the XJ specimen exhibits better linear elasticity in the initial loading stage. The hysteresis loop is close to a straight line, and the energy absorption is minimal. With the increase in the loading displacement, the cracks gradually expand, the residual deformation increases, and the signs of fusiform hysteresis loops gradually appear. When the load corresponds to a lateral displacement of 43.5 mm, the curve shows a clear yield platform segment corresponding to the specimen yielded. The curve has begun to pinch, showing a bow-shaped hysteresis loop, mainly due to cracks in the lower part of the beam end and small amounts of concrete being crushed and falling. The curves of the three cycles of each stage have a high degree of coincidence when the load corresponds to a lateral displacement of 58 mm. This indicates that the component has a strong recovery ability and has little effect on the performance of the component in repeating loading. When the load corresponds to a lateral displacement of 79.8 mm, due to the spalling of the precast beam concrete, the restoration performance of the members is significantly reduced. At the level of 123.3 mm, the pinch effect of the curve is more obvious, showing a certain inverse S-shape, and the lower longitudinal bars show obvious buckling and slipping. The load reaches 80% of the peak load after a load cycle, and the specimen is damaged. In general, the hysteresis curve is full, corresponding to good energy absorption.(2)It can be seen in the YZ1 specimen that the hysteresis loop changes similar to that of the XJ specimen. However, the hysteresis capability of the YZ1 at the level of 101.5 mm is still strong. Compared with the XJ specimen, the yield load and ultimate load are similar, and the hysteresis curves are identical. However, the pinch phenomenon of the YZ1 specimen was better than that of the XJ specimen at 123.3 mm. The hysteresis and ultimate load-carrying capacity of the YZ1 specimen were stronger than those of the XJ specimen.(3)The YZ2 specimen uses steel strands as longitudinal bars. The fullness of the hysteresis curve is not as good as that of the XJ specimen. The hysteresis loop begins to shrink into an inverse S shape at the 21.8 mm level. From the 43.5 mm level, the recovery performance of the YZ2 specimen begins to decline. It is damaged at the level of 101.5 mm, indicating that YZ2 has insufficient bearing capacity under large displacements. In general, compared with the XJ, the YZ series has a weaker energy absorption capacity and seismic performance.

### 3.3. Stiffness Degradation

Stiffness degradation reflects the cumulative damage of the component under repeated loading, and it is one of the important indicators to analyze the dynamic structural characteristics. The stiffness calculation formula is as follows, and the stiffness regression curve is shown in Figure 12.
(1)$$K_{p}=\frac{\left|+F_{i}\right|+\left|-F\right|}{\left|\Delta_{i}\right|+\left|-\Delta_{i}\right|}$$

It can be found in Figure 12 that the initial rigidity of the XJ specimen is slightly higher than that of the YZ series. This indicates that the existence of seams between the precast components weakens the overall rigidity of the precast specimens. After cracking, the stiffness of the cast-in-place specimen and the precast specimen are the same. All the specimens are in the range of the sixth cycle load to the twentieth cycle load (ranging from a drift ratio of 0.25% to 1%), and the stiffness value has slightly increased. This is due to the accuracy of the rotating support and the fixed base at the bottom. Friction occurs during the rotation at a drift ratio of 0.25% to 1%. Therefore, the stiffness in this range increases due to friction. After the drift ratio reaches 1%, all the specimens enter the yielding state. Although the total reinforcement of the precast specimens is low, the stiffness of the YZ1 after yielding is the same as that of the XJ. The stiffness of the YZ2 after yielding is higher than that of the XJ, indicating that the core beam in the keyway and the configuration of the high-strength tendons can increase the stiffness of the joint after yielding to a certain extent.

### 3.4. Load and Deformation Capacity

The equivalent area method has a relatively clear physical meaning and is widely accepted to determine the yield point [27]. In this research, the area of the equivalent two-fold line and the original curve envelope are equal. The point on the original curve corresponding to the turning point of the equivalent bi-linear model is considered to be the yield point. The schematic diagram is shown in Figure 13.

The yield strength and peak strength are summarized in Table 3. The average yield strengths of the YZ1 and YZ2 are 6.5% and 9% higher than that of the XJ. The average peak strengths of the YZ1 and YZ2 are 5.7% and 14.5% higher than that of the XJ. The yield strength and peak strength of the precast specimens are improved to a certain extent relative to the XJ, indicating that the precast components are not weaker than the cast-in-place components in terms of component strength. The yield ratio of the YZ1 is equivalent to that of the XJ. The YZ2 has a higher yield ratio than the XJ, which indicates that the precast nodes have a relatively high strength safety reserve.

The displacement ductility ratio μ is defined as the ratio of the ultimate displacement to the yield displacement of the joint; the displacement ductility ratio μ in the tensile and push directions are calculated, respectively.
(2)$$\mu =\frac{\Delta_{u}}{\Delta_{y}}$$

The displacement ductility ratio is summarized in Table 4. The ultimate displacement and displacement ductility ratio of the YZ1 with small-diameter and high-strength tendons are larger than those of the XJ. This indicates that the specimens configured with HTRB600 rebars are stronger than the XJ in terms of the ultimate bearing capacity and deformation capacity.

### 3.5. Energy Absorption

The equivalent viscous damping coefficient is an important indicator for measuring the energy absorption capacity of structures for earthquake resistance [28]. The calculation method is shown in Figure 14 and Equation (1).
(3)$$h_{e}=\frac{1}{2\pi }\frac{S_{ABC}+S_{CDA}}{S_{OBE}+S_{ODF}}$$

It can be seen in Figure 15 that there is little difference among the equivalent viscosity coefficients before yielding. After yielding, the viscosity coefficient increased significantly, indicating that energy absorption itself increased significantly. The hysteresis curve was pinched, and the equivalent viscosity coefficient decreased when it was close to failure. After yielding, in the early and middle stages of loading, the viscosity coefficient of the cast-in-place specimen was slightly higher than that of the precast specimens. The precast specimens were inferior to the cast-in-place component at this stage, indicating that the existence of the joints between the precast components caused a slight reduction in energy absorption. However, the equivalent viscosity coefficient of the YZ1 was higher than that of cast-in-place members when it was close to failure, which indicated that the precast specimens could surpass the cast-in-place members when they were close to failure. The YZ2 used steel strands as longitudinally stressed rebars, which had high strength and no obvious yield platform, so the energy absorption capacity was weak.

## 4. Analytical Study of the Beam-Column Joints

### 4.1. Constitutive Model of the Concrete Material and Steel Bar 

Concrete was modeled by the C3D8R element (eight-node three-dimensional solid reduction integral element), and concrete was modeled by concrete constitutive provided by the code for the design of concrete structures (GB 50010, 2010) [29]. The specific formulas are as follows. Figure 16 shows the corresponding uniaxial tensile/compressive stress–strain curves of concrete.

Uniaxial tensile constitutive of concrete.
(4)$$\sigma =\left(1-d_{t}\right)E_{c}\varepsilon \;d_{t}=\{\begin{array}{l} 1-\rho_{t}\left[1.2-0.2x^{5}\right]\;x\leq 1\\ 1-\frac{\rho_{c}}{\alpha_{t}{\left(x-1\right)}^{1.7}+x}\;\;x>1 \end{array}\;x=\frac{\varepsilon }{\varepsilon_{t,r}}\;\rho_{t}=\frac{f_{t,r}}{E_{c}\varepsilon_{t,r}}$$

Uniaxial compression constitutive of concrete.
(5)$$\sigma =\left(1-d_{c}\right)E_{c}\varepsilon \;d_{c}=\{\begin{array}{l} 1-\frac{\rho_{c}n}{n-1+x^{n}}\;\;x\leq 1\\ 1-\frac{\rho_{c}}{\alpha_{c}{\left(x-1\right)}^{2}+x}\;x>1 \end{array}\;\rho_{c}=\frac{f_{c,r}}{E_{c}\varepsilon_{c,r}}\;n=\frac{E_{c}\varepsilon_{c,r}}{E_{c}\varepsilon_{c,r}-f_{c,r}}\;x=\frac{\varepsilon }{\varepsilon_{c,r}}$$

The strength of concrete used in this paper is C40, and the tensile-compression stress-strain curve is shown in Figure 16.

The constitutive model of rebars was shown in Figure 17 [30], OAB and OCD were the tension and compression envelopes of the steel bars, respectively, MN and KL were the unloading sections, LM and NK were the reload sections; M and K were the historical maximum tensile and compressive stress and strain points. Use $\sigma_{m}^{+}$ to indicate the maximum tensile stress, $\sigma_{m}^{-}$ to indicate the maximum compressive stress, $\varepsilon_{m}^{+}$ to indicate the maximum tensile strain, and $\varepsilon_{m}^{-}$ to indicate the maximum compressive strain. The intersection points of the tension and compression unloading section and the strain coordinate axis were represented by $\varepsilon_{0}^{+}$ and $\varepsilon_{0}^{-}$, respectively. The α was called the influence coefficient of hysteretic energy consumption; K_s_ was the initial stiffness; *K_sh_* was the hardening stiffness; *K_sr_* was the unloading stiffness.

Set β = (ε_m −_ ε_0_)/ε_y_, the literature (Vecchio, 1999) stipulates that the unloading stiffness varies according to the size of β; that is, the unloading stiffness is calculated according to the following formula.
(6)$$K_{sr}=\{\begin{array}{cc} K_{s} & \beta <1\\ \left(1.05-0.05\beta \right)K_{s} & 1\leq \beta \leq 4\\ 0.85K_{s} & \beta >4 \end{array}$$

To be able to reflect the characteristics that the steel bars are weak first and then strong under the action of low-cycle repeated loads, a cubic curve is selected.

Such a cubic curve is monotonically increasing in the LM interval, and the cubic function will not increase the difficulty of determining the model. The model loading path formula in this paper is as follows:(7)$$\sigma =\gamma \left(\varepsilon^{3}-\varepsilon^{2}\right)+\left(1-\alpha \right)\sigma_{m}\varepsilon +\alpha \sigma_{m}$$
where $\gamma =K_{sh}\left(\varepsilon_{m}-\varepsilon_{L}\right)-\left(1-\alpha \right)\sigma_{m}$, $\varepsilon =\left(\varepsilon -\varepsilon_{L}\right)/\left(\varepsilon_{m}-\varepsilon_{L}\right)$.

The influence coefficient of hysteretic energy consumption is α, this paper also chooses it to change according to the size of β, and the specific rules are as follows [31]:(8)$$\alpha =\{(\begin{array}{cc} 0.5 & \beta \leq 1\\ 20-\beta )/38 & 1<\beta <20\\ 0 & \beta \geq 1 \end{array}$$

### 4.2. Interface Simulation

Combined with the stress characteristics of the concrete materials, and ignoring the tensile strength of the concrete and the interface bonding force, the contact interface between the precast concrete and the cast-in-place concrete is mainly considered to bear the normal extrusion force and tangential friction force. A variety of interface simulation methods are provided in the ABAQUS software, which can conveniently and quickly simulate the normal extrusion force and tangential friction force of the concrete interface. First, the characteristics of the contact need to be created, and the mechanical properties of the contact surface can be simulated by defining the tangential and normal behavior. For the tangential behavior, this study utilizes the “penalty” friction function with a friction coefficient of 0.6, which is taken as the friction coefficient between the rough concrete surfaces. For normal behavior, this study utilizes hard contact to consider the problem of contact pressure interference and to allow for separation after contact. If the gap between the interface of the precast concrete and the cast-in-place concrete is 0, the contact constraint is achieved through extrusion force. If the gap is less than 0, the two contact surfaces will be separated from each other due to the software parameter settings. Then, the created contact attributes are assigned to the corresponding interface through surface-to-surface contact simulation.

### 4.3. Establishment of the Finite Element Model

Through the above steps, a finite element model consistent with the experimental model was established, and the concrete element and the reinforced element were extracted. The finite element model is shown in Figure 18.

The comparison between the finite element calculation results and the test results is shown in Figure 19. It could be seen that the lateral load calculated by the finite element model was slightly larger than the lateral load obtained from the test results, and the error was controlled within 5–10%. In the early stage of loading, the initial stiffness of the node is larger than the experimental results because the boundary constraint conditions of the simulated node were considered ideally. There was little effect on the later results. The later finite element calculation results agreed well with the experimental results. In general, the method of node model establishment in this paper could accurately predict the mechanical performance of nodes under low-cycle repeated loads and provide a certain theoretical reference for engineering practice.

Figure 20 shows the equivalent plastic strain nephogram of precast joints. Compared with the experimental results, it is seen that the strain nephogram calculated by ABAQUS can reflect the final failure mode of the joints, which is in good agreement with the experimental results.

## 5. Discussion of the Plastic Hinge

The plastic hinge area is a concentrated area where the reinforced concrete components are damaged. In the process of resisting seismic shock waves, more seismic energy will be absorbed and consumed, which is related to the seismic energy dissipation capacity of the entire structure. At the same time, the plastic hinge is also a concentrated area of plastic deformation of components. The ultimate deformation capacity and ductility of the structure are often determined by the deformation capacity of the plastic hinge area. If the deformation capacity of the plastic hinge area is weak, the deformation and ductility of the entire structure under earthquake action will be relatively weak. For post-cast monolithic precast concrete structures, the formation and performance of plastic hinges play a more important role in the seismic performance of the entire structure. Many precast concrete structures are due to the insufficient formation of plastic hinges under the action of earthquakes, resulting in weaker seismic performance than cast-in-place concrete structures.

The existing experimental results show that relocating beam plastic hinge zones away from the column face can be achieved by using appropriate strengthening methods so that the concrete oblique cracks and steel bond slip are significantly reduced. Therefore, the seismic performance of beam-column joints can be greatly improved [32,33,34]. In this test, it was found through the destruction of 3 nodes that because the steel strands at the bottom of the precast beam extend into the opposite beam at the YZ2 node, the strength of the joint surface of the beam and column was enhanced, which could promote the plastic hinge of the prefabricated beam end to move outward. As shown in Figure 21, it could be seen that the plastic hinge area of the YZ2 node was significantly outwardly moved. This design could effectively prevent the plastic hinge from developing into the panel zone and avoid damage to the entire structure due to the shear failure of the panel zone. In the later experimental research, reasonable design measures could be taken to enhance the strength of the beam-column junction, so that the plastic hinge of the beam end could be kept away from the beam-column panel zone and a set of corresponding design methods could be formed.

## 6. Conclusions

This study has improved the connection based on previous research. Also, the seismic performance of the connections was studied, and the following conclusions were obtained through the research.
(1)The ultimate displacement and ductility coefficient of the YZ1 was higher than those of the cast-in-place node, which indicated that the ductility of the YZ1 was not lower than that of the XJ and ensures that the YZ1 had better ductility under the action of seismic.(2)YZ2 shows insufficient ductility and energy consumption. When the component is damaged, the deformation of the ordinary additional steel bar is small, so the seismic performance is not as good as that of the cast-in-place component.(3)Using the steel hysteresis model proposed in this article, the mechanical performance of precast concrete beam-column joints can be simulated more accurately, which provides a theoretical reference for the further improvement and design of joints.(4)The strength of the beam-column joint could be enhanced by adding additional steel bars, dampers, and pre-embedded section steel at the beam-column joints to promote the plastic hinges moving outward to the beam ends, and this could help to avoid the collapse of the entire structure due to the development of the plastic hinge in the panel zone.

## Figures and Tables

**Figure 1 materials-15-07127-f001:**
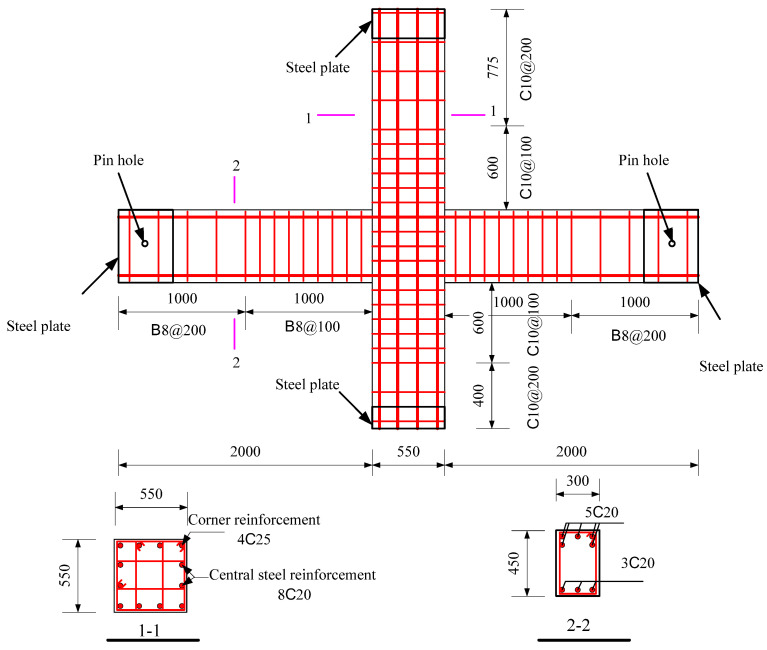
The design details of XJ (Unit: mm). Note: B = HRB335, HRB335 represents steel bars with a strength grade of 335 MPa; C = HRB400, HRB400 represents steel bars with a strength grade of 400 MPa.

**Figure 2 materials-15-07127-f002:**
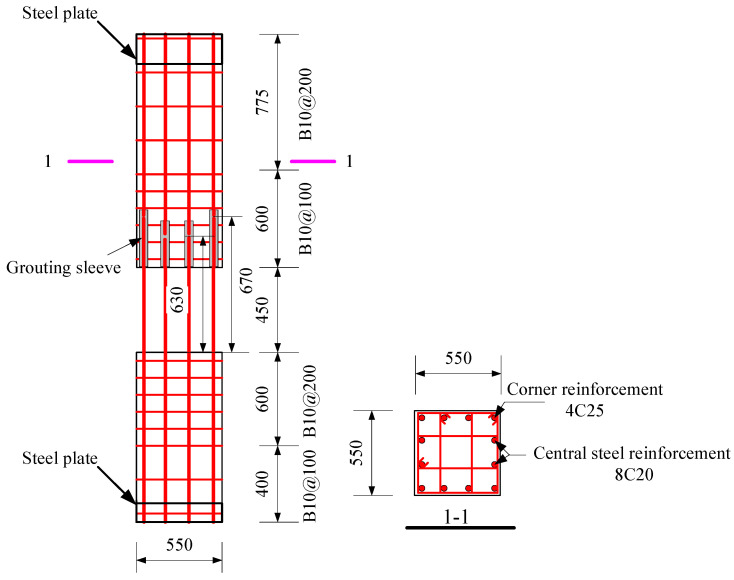
A detailed design of the precast columns (Unit: mm).

**Figure 3 materials-15-07127-f003:**
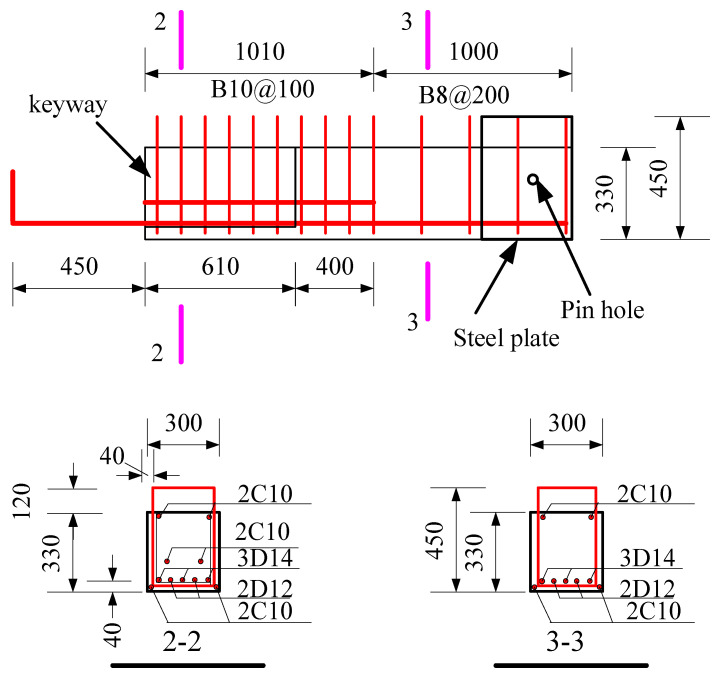
The precast beam details of YZ1 (Unit: mm). Note: D = HTRB600, HTRB600 represents steel bars with a strength grade of 600 MPa.

**Figure 4 materials-15-07127-f004:**
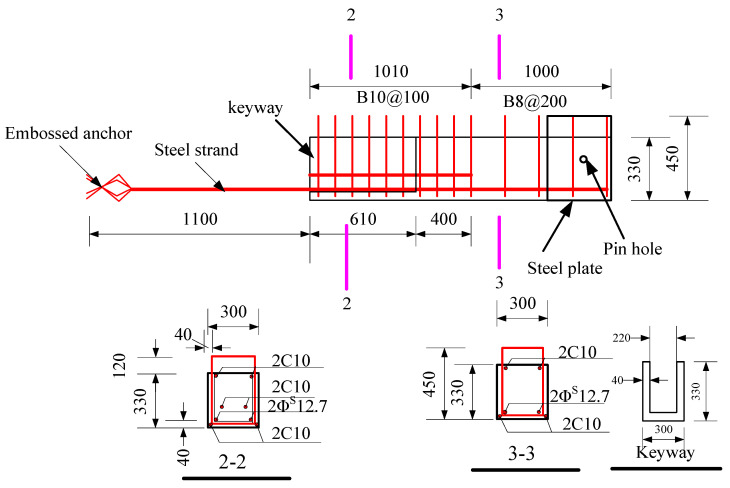
The precast beam details of YZ2 (Unit: mm).

**Figure 5 materials-15-07127-f005:**
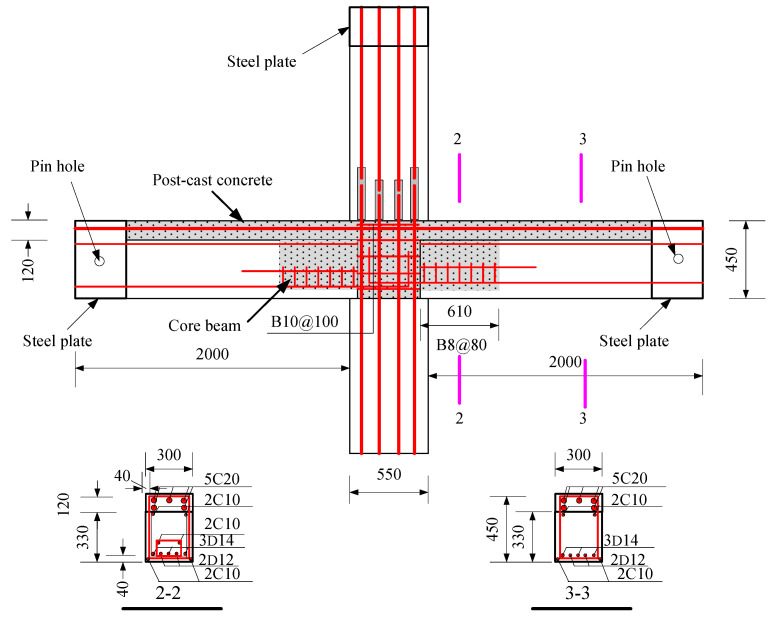
The assembly design details of YZ1 (Unit: mm).

**Figure 6 materials-15-07127-f006:**
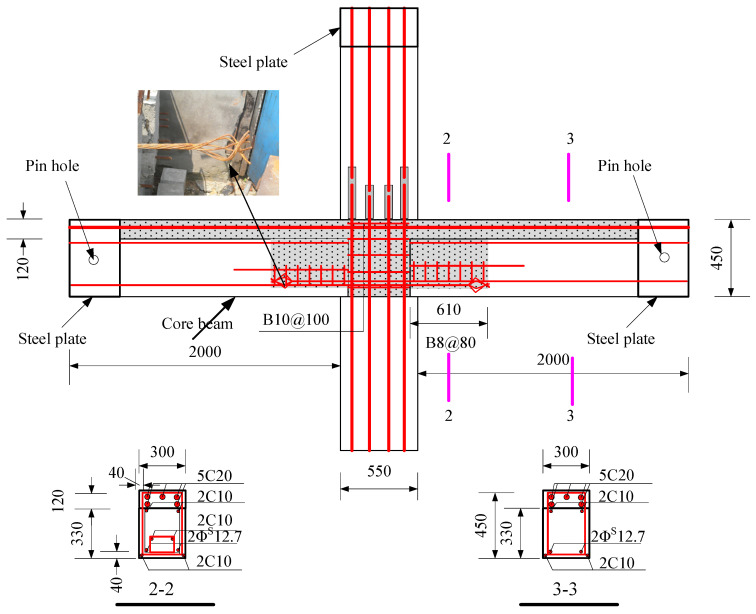
The assembly design details of YZ2 (Unit: mm).

**Figure 7 materials-15-07127-f007:**
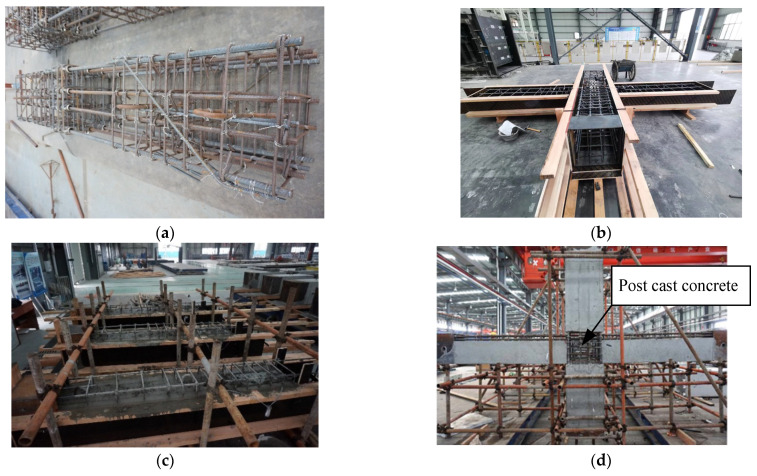
The specimens preparation. (**a**) Reinforcing cage for the column. (**b**) Formwork support. (**c**) Pouring concrete. (**d**) Assembly.

**Figure 8 materials-15-07127-f008:**
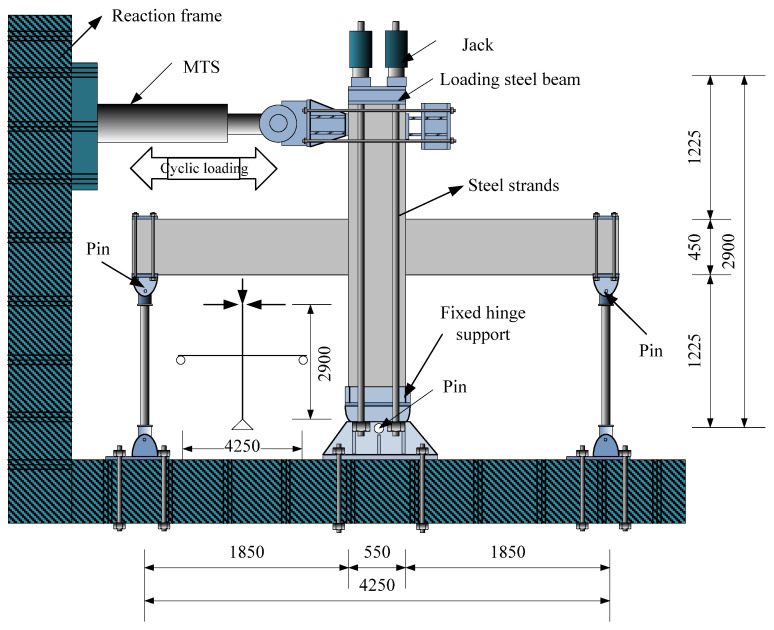
The loading device (unit: mm).

**Figure 9 materials-15-07127-f009:**
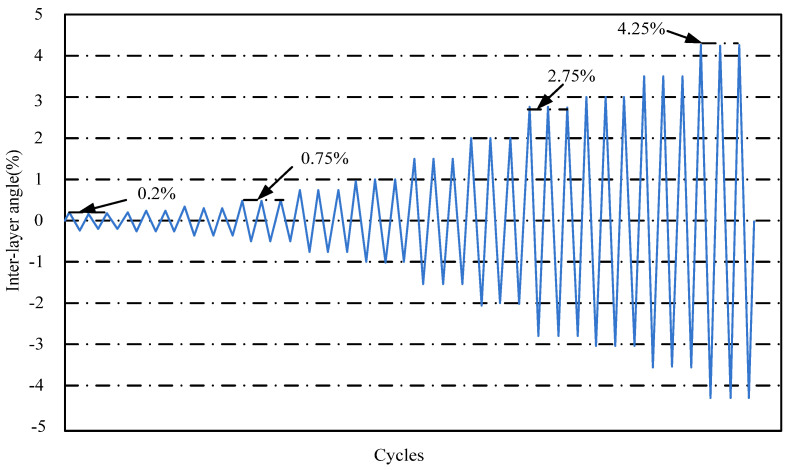
The loading method.

**Figure 10 materials-15-07127-f010:**
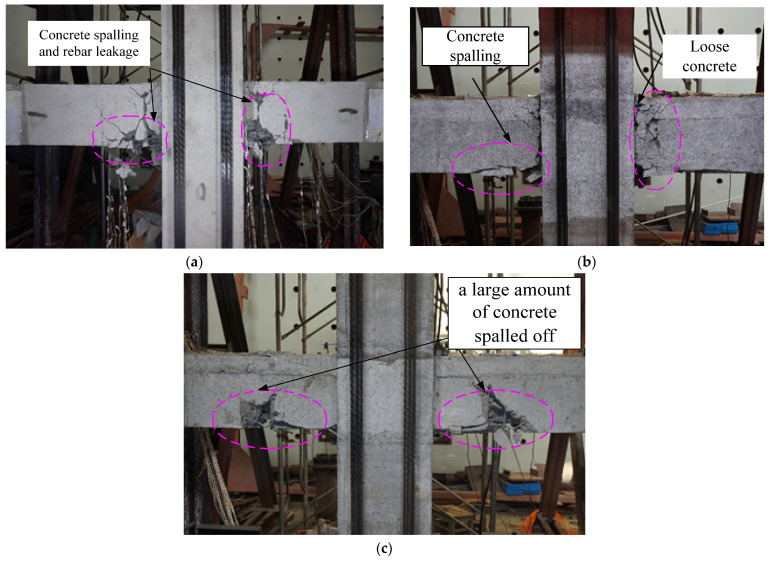
The final destruction pattern of the joints. (**a**) XJ. (**b**) YZ1. (**c**) YZ2.

**Figure 11 materials-15-07127-f011:**
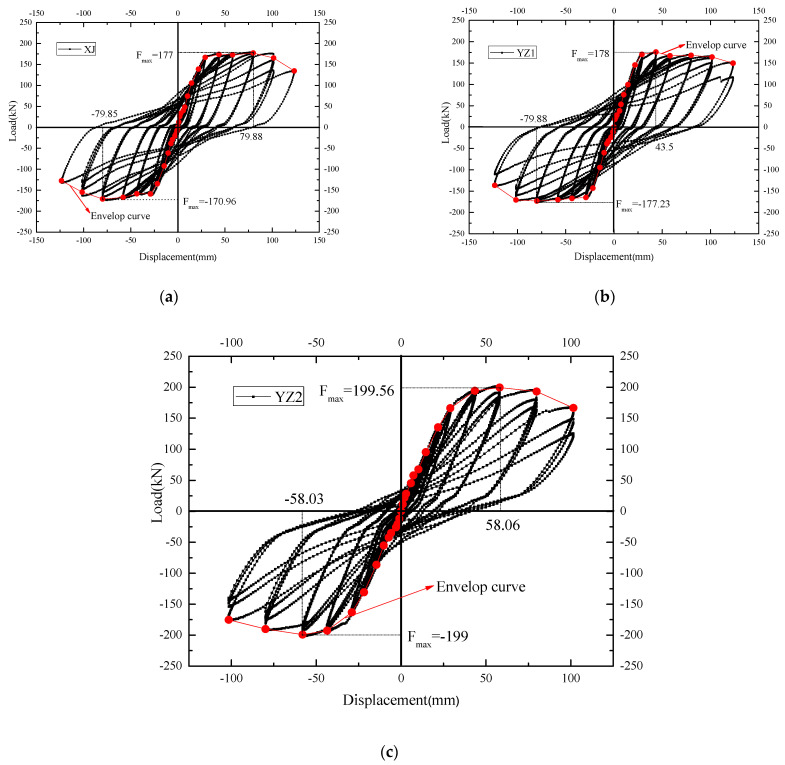
The hysteresis curve and skeleton curve of the joints. (**a**) XJ. (**b**) YZ1. (**c**) YZ2.

**Figure 12 materials-15-07127-f012:**
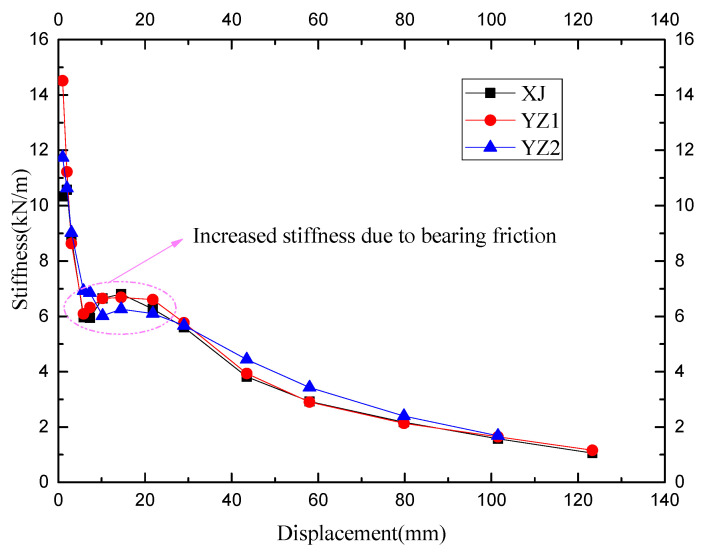
The stiffness degradation.

**Figure 13 materials-15-07127-f013:**
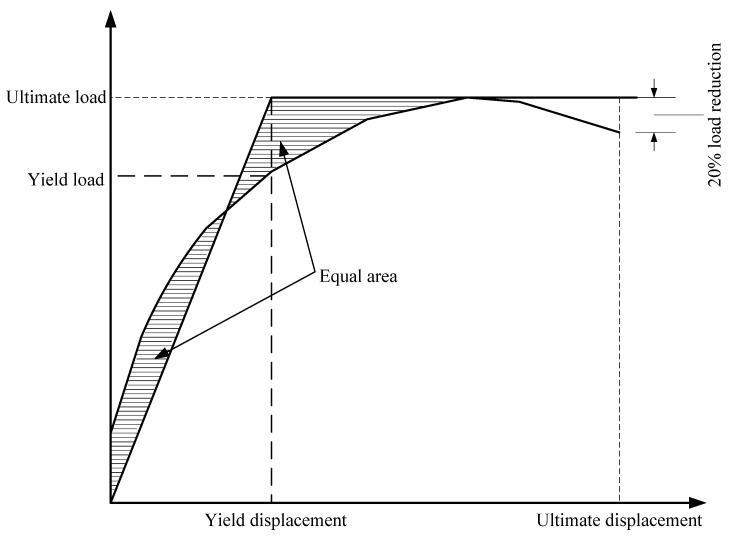
A schematic diagram of the equal area method.

**Figure 14 materials-15-07127-f014:**
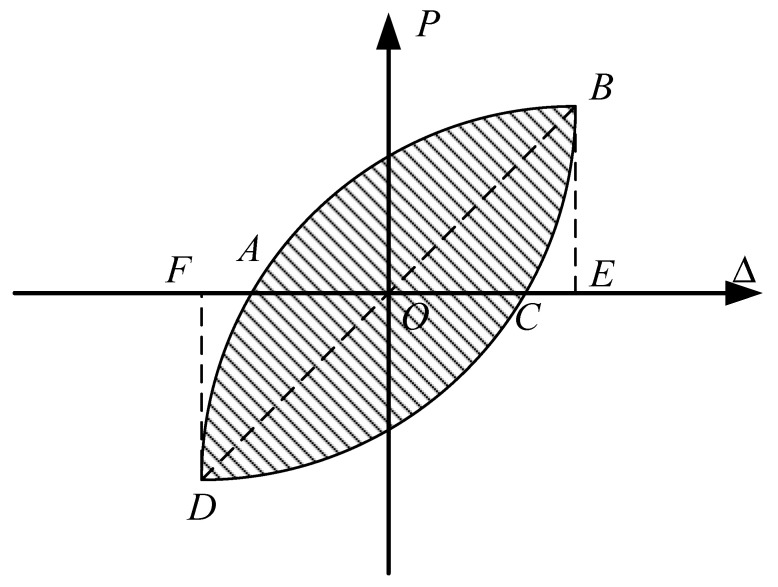
A schematic diagram of the calculation of equivalent viscosity coefficient.

**Figure 15 materials-15-07127-f015:**
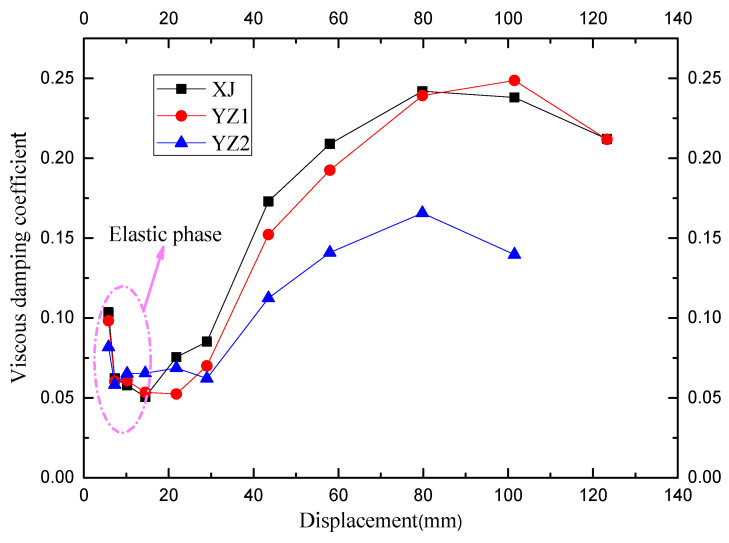
A comparison of the equivalent viscous damping coefficients.

**Figure 16 materials-15-07127-f016:**
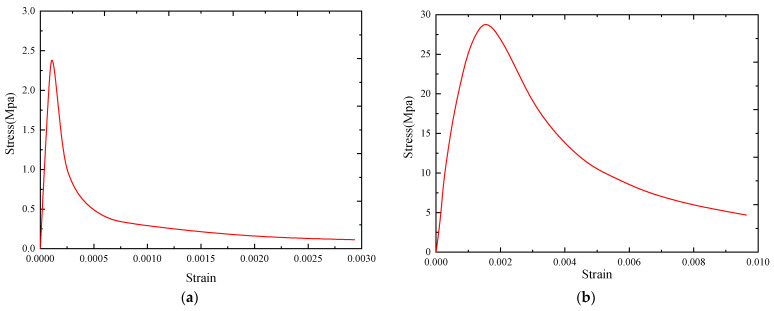
The tension and compression stress-strain curve of concrete. (**a**) Concrete uniaxial tensile stress-strain. (**b**) Concrete uniaxial compressive stress-strain.

**Figure 17 materials-15-07127-f017:**
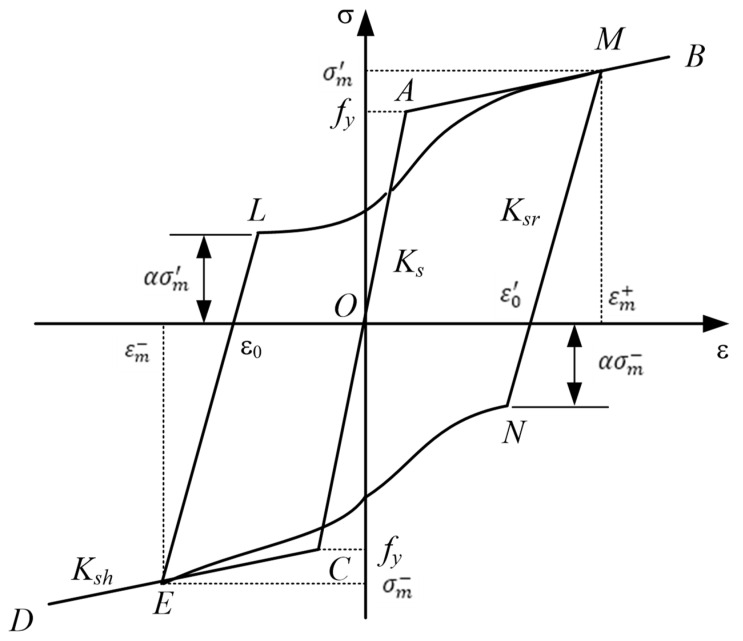
A constitutive model of the rebars.

**Figure 18 materials-15-07127-f018:**
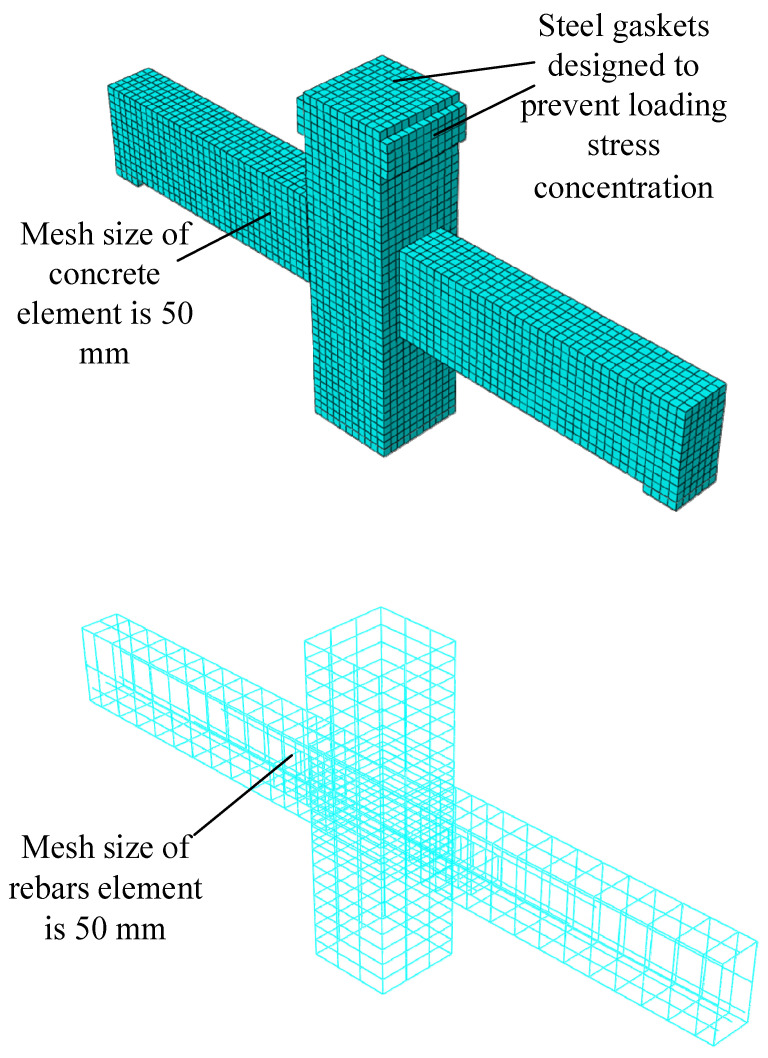
The finite element model.

**Figure 19 materials-15-07127-f019:**
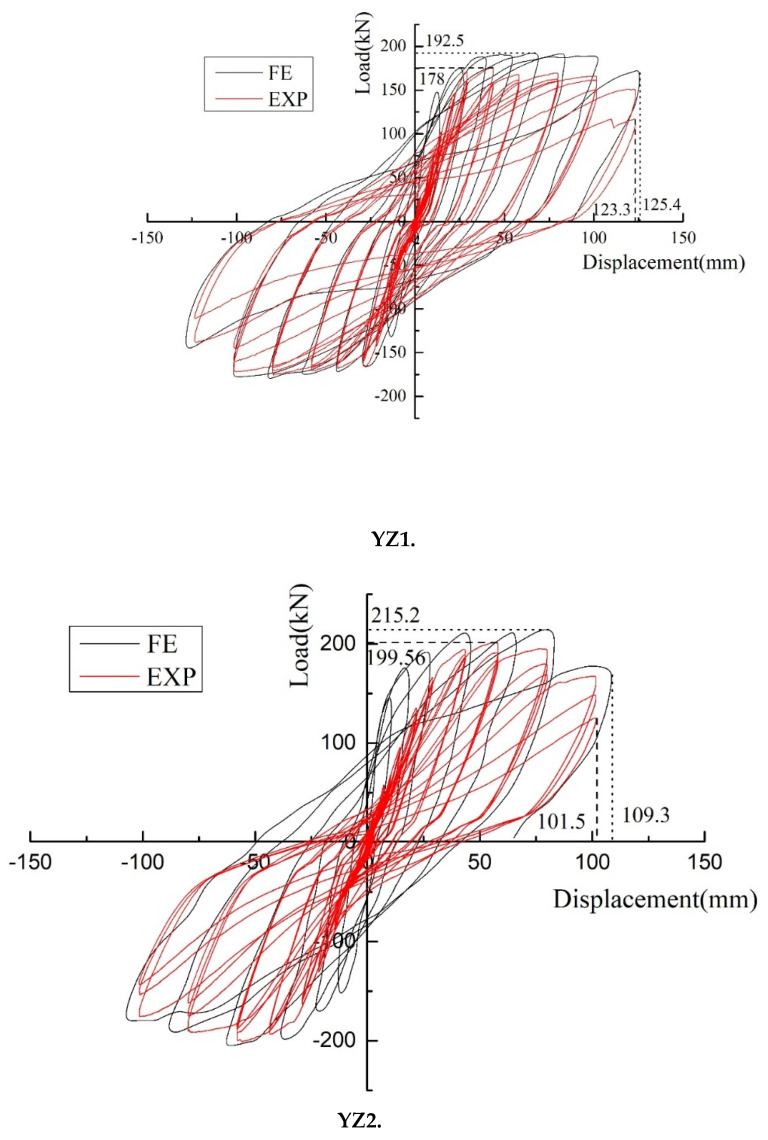
A comparison of finite element calculation and test results.

**Figure 20 materials-15-07127-f020:**
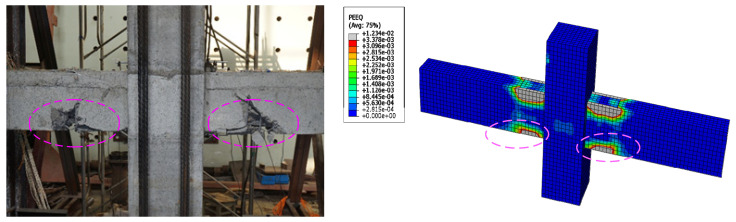
The failure morphology.

**Figure 21 materials-15-07127-f021:**
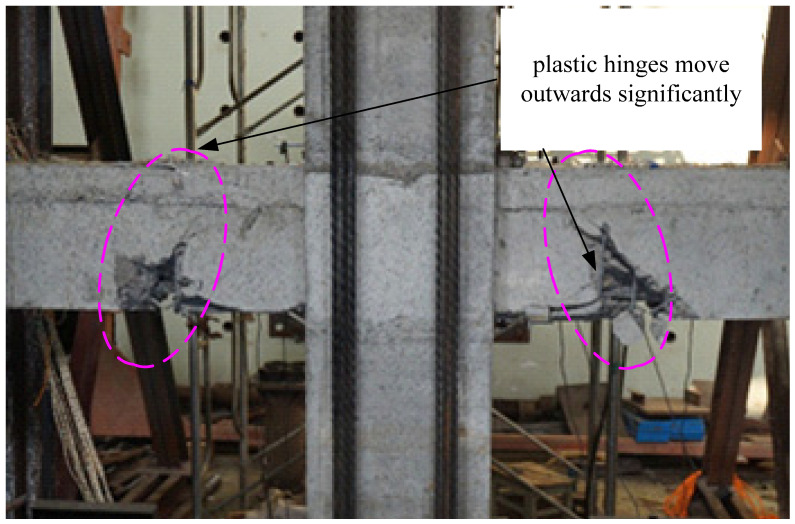
The plastic hinges moving outward on YZ2.

**Table 1 materials-15-07127-t001:** The mechanical properties of steel bars.

Steel Grade	Diameter/mm	Area/mm ^2^	Yield Strength *f_y_*/MPa	Tensile Strength *f_u_*/MPa	Elastic Modulus *E_s_*/MPa
HRB400	10	78.5	420.1	619.3	2.0 × 10^5^
HRB400	14	201.1	411.2	627.1	2.0 × 10^5^
HRB400	20	314.2	425.4	617.2	2.0 × 10^5^
HRB400	22	380.1	420.8	617.4	2.0 × 10^5^
HRB400	25	490.9	423.7	610.0	2.0 × 10^5^
Strand	12.7	98.7	1620	1927	1.95 × 10^5^
HTRB600	12	113.1	636	773	2.0 × 10^5^
HTRB600	14	201.1	631	813	2.0 × 10^5^

**Table 2 materials-15-07127-t002:** The mechanical properties of concrete.

Concrete Batch	Axial Tensile Strength/MPa	Axial Compressive Strength/MPa	Elastic Modulus *E_c_*/MPa
First batch	2.75	35.8	3.22 × 10^4^
Second batch	2.77	36.2	3.41 × 10^4^

**Table 3 materials-15-07127-t003:** The carrying capacity of the joints.

Node	Direction	Yield Load (kN)	Peak load (kN)	Strong Yield Ratio	Average Value
XJ	forward	166.55	177	1.06	1.07
	reverse	−158.87	−170.96	1.08	
YZ1	forward	173.32	178	1.03	1.02
	reverse	−173.20	−177.23	1.02	
YZ2	forward	176.33	199.56	1.13	1.12
	reverse	−178.51	−199	1.11	

**Table 4 materials-15-07127-t004:** The deformation capacity.

Specimen	Direction	Yield Displacement (mm)	Ultimate Displacement (mm)	Ductility Factor	Average Value
XJ	Forward	28.99	118.16	4.08	3.82
	Backward	−32.53	−115.97	3.57	
YZ1	Forward	32.75	123.30	3.76	3.97
	Backward	−29.55	−123.30	4.17	
YZ2	Forward	34.29	101.50	2.96	2.87
	Backward	−36.61	−101.50	2.77	

## Data Availability

Not applicable.

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
