# Peer review of "Seismic Performance of Precast Concrete Frame Beam-Column Connections with High-Strength Bars"

_materials, 2022, doi:10.3390/ma15207127_

Round 1

Reviewer 1 Report

The topic presented in the paper is an interesting one with the construction industry gradually shifting back from cast-in-place structures to prefabricated ones.

There are some issues that need addressing before considering this manuscript for possible publication.

Abstract - lines 3-5 - the sentence needs to be re-written as it has no beginning and no end.

I am not sure the authors developed a FEA method but a FEA model to reproduce the experiment

Keywords - seismic? Please consider "seismic behavior" or "lateral cyclic loading"

Page 2, lines 18-22 - there are 2 sentences in a single statement. The authors should separate the two right where the comma is used.

Page 2, line 43 - what do you understand by "fortified earthquake"?

Page 3, line 7 - why do the authors think high-strength steel is better than the normal steel? It is well known that high strength steel has a much smaller yielding plateau, sometimes it misses completely, compared to the normal strength steel. Did you measure the rebar strain during the experiment? Did rebars yielded or even fractured? If smaller diameter bars were used, one would need more bars to cover the width of the beam since each rebar has its own "area of influence". Last but not least, high strength steel requires more energy to be produced, so the justification of emission reduction does not really stand its ground.

Page 3, Section 2.1 - please substitute "losed" by "lost"

The authors need to first explain the meaning of letters B, C and D used for rebars and stirrups. These notations are somehow explained in Figure 1 but it is still difficult to understand what they are referring to.

The geometry of the keyway should be presented in either Figure 3 or Figure 4 or in both if the geometry was different.

What is the meaning of DH, with H being the superscript, in Figure 3 (Sections 2-2 and 3-3)? The same notation repeats itself in Figure 5.

Page 7 - what do you understand by laminated concrete?

Figure 7a - reinforcing cage for the column

Figure 7b - does the foreground of the photo show one beam end or column end?

Figure 7d - please provide a more clear photo

Page 7, Section 2-3 - I think the authors refer to "direct tensile test" instead of "pull out", unless the intention was to determine the bond strength (nowhere else mentioned in the paper).

Page 8, section 2.4 - "designed axial compressive strength" or better "axial load carrying capacity" if referred to the column.

Pages 10, 11 - the load can not reach 43.5, 58 or 79.8 mm. "The load corresponding to a lateral displacement of...."

Page 11 - "the load reaches 80%..."

Page 13, section 2.4 - what do you mean by "two-fold line"? Are the authors referring to "bi-linear model"?

Figure 16 is not cites anywhere in the text (please mention the source unless it is authors' work)

Section 4.1 - pleas remove the "constitutive" from the end of section title

Figure 18 is wrongly cited as it should be Figure 16

Equations are written in much larger fonts

Page 20 - what do you understand by "extrusion force"?

Figure 20 - which analysis case is the figure presenting?

Page 22 - "as shown in Figure 21..."

Please check the quality of English language, too. Some formulations are hard to understand.

Reviewer 2 Report

Comments are listed below:
1. Strengthen the abstract section. Add the key conclusion of the works in the last two lines of the abstract section. Remove the unnecessary information.
2. Discuss the novelty of the work in respect of the application 
3. There are numerous spelling and grammatical errors. Please revise the manuscript thoroughly. Sentences are also not complete and references are also cited in a rough manner. 
4. Try to make a bridge between current and previously published work and specify the gap area and objective of the work. The introduction section is very poor: refer to following published work: 
- Wenchen Ma. (2021). “Behavior of Aged Reinforced Concrete Columns under High Sustained Concentric and Eccentric Loads,” Doctoral Thesis, University of Nevada, Las Vegas, Nevada, USA, 198 p.

5. Provide the image of the experimental setup with good quality. Also, add the image of the welded pipe produced.
6. The results are ok but the discussion section is very poor. It looks like a technical report instead of a technical article. Improve the discussion section and add more references in support of the results.
7. Shorten the length of the conclusion section.
8. The work is good, but the technical discussion and introduction section needs improvement. Paper can be accepted after following minor corrections.

Round 2

Reviewer 1 Report

The revised manuscript is in much better shape than the original submission. The authors have addressed the issues raised during the reviewing process.

In light of the new changes, I endorse the manuscript for publication.

Author Response

Thank you very much for the reviewer 's recognition of the work

Reviewer 2 Report

You should consider these comments again:

1. There are numerous spelling and grammatical errors. Please revise the manuscript thoroughly. Sentences are also not complete and references are also cited in a rough manner. 
2. Try to make a bridge between current and previously published work and specify the gap area and objective of the work. The introduction section is very poor: refer to following published work: 
- Wenchen Ma. (2021). “Behavior of Aged Reinforced Concrete Columns under High Sustained Concentric and Eccentric Loads,” Doctoral Thesis, University of Nevada, Las Vegas, Nevada, USA, 198 p.
